# WO_3_ Nanopores Array Modified by Au Trisoctahedral NPs: Formation, Characterization and SERS Application

**DOI:** 10.3390/ma15238706

**Published:** 2022-12-06

**Authors:** Jan Krajczewski, Robert Ambroziak, Sylwia Turczyniak-Surdacka, Małgorzata Dziubałtowska

**Affiliations:** 1Faculty of Chemistry, University of Warsaw, 1 Pasteur St., 02-093 Warsaw, Poland; 2Institute of Physical Chemistry, Polish Academy of Sciences, Kasprzaka 44/52, 01-224 Warsaw, Poland; 3Biological and Chemical Research Centre, Faculty of Chemistry, University of Warsaw, 101 Żwirki i Wigury Street, 20-089 Warsaw, Poland

**Keywords:** SERS, plasmonics, WO_3_ nanostructures, Au TOH NPs, anodization

## Abstract

The WO_3_ nanopores array was obtained by an anodization method in aqueous solution with addition of F^-^ ions. Several factors affecting the final morphology of the samples were tested such as potential, time, and F^-^ concentrations. The morphology of the formed nanopores arrays was examined by SEM microscopy. It was found that the optimal time of anodization process is in the range of 0.5–1 h. The nanopores size increased with the increasing potential. The XPS measurements do not show any contamination by F^-^ on the surface, which is common for WO_x_ samples formed by an anodization method. Such a layer was successfully modified by anisotropic gold trisoctahedral NPs of various sizes. The Au NPs were obtained by seed-mediated growth method. The shape and size of Au NPs was analysed by TEM microscopy and optical properties by UV-VIS spectroscopy. It was found that the WO_3_-Au platform has excellent SERS activity. The R6G molecules could be detected even in the range of 10^−9^ M.

## 1. Introduction

SERS spectroscopy (Surface-Enhanced Raman Scattering) is a very useful analytical tool due to its high precision, non-destructive character, and low detection limit [1]. The typical SERS spectrum is a fingerprint of molecules that allows for analyzing even complicated molecules as well as DNA, RNA, or peptide strands [1]. There are some literature reports about even single molecule detection by the SERS method [1].

The development of a novel SERS platform with excellent SERS properties is still desired. The ideal SERS platform should exhibit high EF (Enhancement Factor) and repeatability from point-to-point analysis as well as from sample-to-sample analysis. Time and chemical stability are other important factors.

The ideal SERS platform should consist of repetitively placing nanostructures over the whole sample in a homogenous way. It is possible to form some nanostructures from pure plasmonic metals by lithography [2] and electrochemistry [3], but the simplest way is to form a nanostructured material from another material and then cover its surface with a plasmonic layer. Such a solution was applied for TiO_2_ [4] and ZrO_2_ NTs (Nanotubes) [5], ZnO nanorods [6], and InP nanowires [7]. However, the main limitation of this method results from the fact that the deposited plasmonic layer flattens the nanostructure’s surface. For example, it is proved that in the ZrO_2_ NTs, the thickest layer of 50 nm covers the morphology of the nanotubes, which leads to a decrease in EF [8]. Therefore, the other possibility is to deposit the plasmonic nanoparticles on the nanostructured surface [9]. The presence of a nanostructured surface leads to the homogenous distribution of NPs even by simple drop casting and prevents the so-called coffee ring effect [10]. Additionally, the application of NPs with an anisotropic shape instead of a typical semi-spherical shape allows the increase of EF and, in consequence, decreases the LOD (limit of detection) for many analysts.

One of the interesting structures are WO_3_ nanopores that exhibit interesting properties. Such media can be used as chemical sensors [11] or in applications where the properties of electrochromism are used [12]. However, the synthesis of highly organized WO_3_ nanopores by electrochemical methods is still under development [13,14,15,16,17]. For example, Schmuki et al. proposed a synthesis method without the use of electrolytes containing fluorides [18]. Although nanopores were obtained, they were closed at the top. Another example of a synthesis method leading to well-formed nanopores involves the use of hot, pure orthophosphoric (V) acid, which is dangerous because there is a risk of chemical and thermal burns, and it is also corrosive [19]. However, one of the papers proposed the electrochemical synthesis of WO_3_ nanostructures with well-formed nanopores. It is a very promising method and we decided to experimentally determine the influence of the parameters on the obtained morphology. So far, WO_3_ nanostructures formed by an anodization method have not been used as a SERS platform, but instead have found an application mainly in the photocatalysis.

In this work, we report about synthesis parameterization of a well-ordered WO_3_ nanopores array by an anodization method in aqueous solution containing F^-^ ions. Parameters like time, F^-^ concentrations, and applied potential affecting the final morphology of the array were carefully examined. The XPS analysis showed that formed WO_3_ nanopores array is not contaminated by F^-^ ions, which is typical for WO_3_ array preparation in organic solvents. After deposition of trisoctahedral Au NPs with various sizes, the nanoarray could be applied as a SERS active platform. It was found that the size of deposited NPs affects the EF.

## 2. Materials and Methods

### 2.1. Materials

Tungsten foil (0.25 mm thick and 99.5% purity) was purchased from Alfa Aesar. NaF, NH_4_F, HF (40%), Ascorbic Acid (AA), CTAB (Cetyltrimethylammonium bromide), CTAC (Cetyltrimethylammonium chloride), NaBH_4_ (Sodium borohydride), and NaOH (Sodium hydroxide) were purchased from Sigma-Aldrich (Darmstadt, Germany). Sulfuric acid (H_2_SO_4_) and isopropanol (99%) were purchased from POCH S.A. A 30% solution of tetrachloroauric(III) acid in diluted HCl was purchased from the Polish State Mint. The water was purified by a Millipore Milli-Q system and had a resistivity of ca. 18 MΩ/cm.

### 2.2. Synthesis of Flat WO_3_ and WO_3_ Nanoporous Array

The tungsten foil was cut by guillotine for metals onto pieces with 6 mm × 12 mm dimensions. The W samples were cleaned by immersing in acetone, ethanol, and water, respectively, and sonicated in each solution for 5 min. Then, W foil was electrochemically polished in 0.25 M NaOH aqueous solution in a two-electrode cell where the W foil was used as the anode and the platinum counter electrode as the cathode for 8 min 30 s with a potential of 8 V.

The anodization was carried out in one step in 1 M H_2_SO_4_ solution containing various amounts of NaF (from 0.02 M to 0.15 M NaF in 100 mL of electrolyte). The potential during the anodization process was constant and was in the 20–60 V range. The anodization was carried out for 60 min. The flat WO_3_ sample was formed in the same condition (40 V) in the absence of fluoride ions.

### 2.3. Synthesis of Au Trisoctahedral Nanoparticles

The trisoctahedral gold nanocrystals were synthesised by modified seed-mediated growth method [20]. In the first step, the small semi-spherical nanoseeds were formed by chemical reduction. For this purpose, 92 µL of 10 mM HAuCl_4_ was added to 7 mL of 75 mM aqueous solution of CTAB. The solution was vigorously stirred and then 0.42 mL of ice-cold NaBH_4_ solution (10 mM) was injected at once. The solution changed colour from yellow to brown, indicating the formation of small, gold nanoparticles. The solution was then aged for 2 to 5 h at 30 °C. The trisoctahedrons with an average size of 70 nm were formed by the regrowth of nanoseeds. The growth solution was prepared by mixing 0.25 mL of 10 mM HAuCl_4_ solution with 9 mL of 22 mM CTAC solution. Then, the freshly prepared ascorbic acid solution (3.06 mL of 38.8 mM) was added. The colour of the solution changed from yellow to colourless. Finally, the 50 µL of 100 times diluted nanoseeds was added and the solution was gently mixed for 10 s. The solution changed colour to intense pink in three minutes. The bigger trisoctahedrons were formed by regrowth of 70 nm trisoctahedrons. All procedures were the same, but instead of adding semi-spherical nanoseeds, the previously formed trisoctahedrons were added.

Finally, the trisoctahedrons were centrifuged to stop the growth process. The supernatant containing nanoparticles was redispersed in the same amount of Millipore water. For SERS measurements, the trisoctahedrons solution was 2 times concentrated by one more centrifugation cycle and redispersed in isopropanol. 60 µL of Au NPs was dropped onto the middle of the WO_3_ surface.

### 2.4. Characterization Methods

The SERS platform was prepared by the simple drop casting of the solution containing Au trisoctahedrons of various sizes on the WO_3_ nanopores surface. In each case, 25 µL of Au trisoctahedrons was dropped four times and then samples were dried at 60 °C in the vacuum. 

The SERS measurements were carried out after dropping 10 µL of R6G (Rhodamine 6G) solution of various concentrations in the range of 10 mM to 100 pM.

The morphology of the samples was examined using a scanning electron microscope (SEM, a FEI Nova NanoSEM 450, Brno, Czech Republic). For imaging, low-energy electron detectors, an Everhart–Thornley detector (ETD) and a through-the-lens (TLD) detector, were used in the low- and high- resolution modes, respectively. All modes were performed in the same configuration at a primary beam energy of 10 kV.

Material cross-section was performed using electron-ion (Ga+) scanning (Crossbeam 540X microscope Jena, Germany). Prior to the cut, the sensitive-to-the-ion-beam surface was protected with electron-beam-induced Pt deposition (2 kV, 4 nA) and after that with ion-beam-induced Pt deposition (30 kV, 300 pA). The cross-section was performed with 30 kV and 3 nA, whereas the final polishing was stopped at 30 kV and 700 pA.

Transmission electron microscopy (TEM) analyses were carried out with a Zeiss LIBRA 120 electron microscope working at an accelerating voltage of 120 kV. The microscope was equipped with an in-column OMEGA filter. The sample of nanoparticles obtained was deposited onto a 300-mesh copper grid coated with a Formvar layer.

UV-VIS spectra were recorded using a Thermo Scientific Evolution 201 spectrophotometer. 

The Raman measurements were carried out using a Horiba Jobin-Yvon Labram HR800 spectrometer equipped with a Peltier-cooled charge-coupled device detector (1024 × 256 pixels), a 600 groove/mm holographic grating, and an Olympus BX40 microscope with a long-distance 50× objective. An He-Ne laser provided the excitation radiation of a 632.8 nm wavelength.

The chemical states of individual elements were verified by X-ray photoelectron spectroscopy (XPS) using a Microlab 350 (Thermo Electron, East Grinstead, UK) spectrometer. For this purpose, the X-ray excitation source (AlKα anode: power 300 W, voltage 15 kV, beam current 20 mA) was used. The lateral resolution of XPS analysis was about 0.2 cm^2^. The high-resolution XPS spectra were recorded using the following parameters: pass energy 40 eV and energy step size 0.1 eV. XPS spectra were reprocessed using the CasaXPS (2.3.18PR1.0) software. Spectra were fitted with GL(30) line shape after Shirley background subtraction and subsequently charge corrected to give a C 1 s at 285 eV.

## 3. Results and Discussion

### 3.1. Structural Characterization

The WO_3_ nanopores array were formed by a standard anodization process in which nanopores were formed perpendicular to the tungsten substrate. During the anodization, the W foil acts as an anode and hence various tungsten oxides are formed. The platinum counter electrode acts as a cathode on which formation of hydrogen occurs. The whole process can be described by the following equations [21]:W + 2H_2_O ↔ WO_2_^2+^ + 4H^+^ + 6e^−^
2WO_2_^2+^ + H_2_O + 2e^−^ ↔ W_2_O_5_ + 2H^+^
W_2_O_5_ + H_2_O ↔ 2WO_3_ + 2H^+^ + 2e^−^

Due to the presence of fluoride ions, WO_3_ form soluble a complex with fluoride ions:WO_3_ + 2H^+^ + 4F^−^ ↔ WO_2_F_4_^2−^ + H_2_O

The first synthesis showed that the commercially available W foils are not flat enough to produce samples containing large areas homogeneously covered by WO_3_ nanopores. Therefore, the tungsten foil was electrochemically polished in an aqueous solution of NaOH. The potential was kept at 8 V and the duration was 8 min 30 s. After that process, SEM (Scanning Electron Microscope) images showed atomically flat W surfaces. 

As in a typical anodization method of formation of nanopores metal oxide nanoarray, many parameters could affect size, distribution, and efficiency of nanostructure formation. It was found that the optimal distance between Pt counter electrode and W foil was 1 cm. With greater distance, the formed nanostructures are deformed and are not evenly distributed over the surface of the W foil. The other important factor is the ratio of the surfaces of both electrodes. The samples with well-defined WO_3_ nanopores are only formed when the Pt counter electrode surface is not smaller than the surface of the W foil.

For many nanostructure materials formed by the anodization process, it has been reported that the electrolyte solution should be aged. It is a common procedure applied in organic solvents containing EG (ethylene glycol) [22,23], glycerol [24], or others [13,25]. This is caused by the weak solubility of the F^-^ source in organic media. The solution must be vigorously stirred for the complete dissolution of fluoride salt. This process is accompanied by the formation of an air bubble in the viscous organic solution. The presence of the air bubble disturbs the migration of F^-^ ions and hence the growth of nanostructures is not homogenous. It was found that there are not any significant changes in the morphology of WO_3_ nanoarray formed with freshly prepared H_2_SO_4_ NaF solution compared to those formed with solution aged for 24 h or for three days. 

The quality of the obtained nanopores largely depends on the composition and conditions of anodization [26]. The influence of the type of electrolyte, of voltage and concentration for the selected electrolyte, and of the anodization time were examined.

All electrolytes used were water electrolytes. Anodization was performed at 40 V for 60 min in the presence of 0.1 M NaF and 0.1 M NH_4_F in a solution of 1 M H_2_SO_4_ and with 0.1 M (NH_4_)_2_SO_4_ in water. Well-developed nanopores were obtained in electrolytes containing NaF and NH_4_F. Only in the (NH_4_)_2_SO_4_ solution was the desired structure not obtained. Based on the work of K. Syrek et al., it was found that structures obtained with the use of (NH_4_)_2_SO_4_ may arise due to the slow formation of pores in WO_3_ [26]. The pore layer growth process is strictly dependent on the presence of factors that can dissolve the WO_3_ layer. If no ions complexing with WO_3_ such as H^+^ or F^-^ are present in the solution, too few holes are formed in the WO_3_ layer to form pores. In the (NH_4_)_2_SO_4_ solution, the pH is slightly acidic, but the amount of H^+^ ions is insufficient to significantly etch the surface of tungsten (VI) oxide, therefore only single pores are visible in further stages of growth.

The best-formed pores are visible in the case of the sample made in NaF solution (Figure 1). It differs from the NH_4_F solution only in the cation. The reason for this is the lower mobility of NH_4_^+^ ions compared to smaller Na^+^ ions. This makes it difficult to dissociate the dissolved substances from the metal surface in the case of samples with NH_4_^+^ cations (Figure 1a–d). The cross-section image (Figure 1b) allows one to determine the thickness of the obtained oxide layer. It is relatively equal over the entire length imaged and amounts to 270 nm.

The very important parameter during the formation of semiconducting nanostructures oxide layer by the anodization method is the time of the anodization process. Figure 2 shows steps of the formation of WO_3_ nanopores into W foil after anodization in 1 M H_2_SO_4_ solution of 0.1 M NaF at various times. These SEM images showed that in the first step, the small, randomly distributed pores are formed. Then, increasing processes lead to the formation of more and more WO_3_ pores, which finally lead to an assembled layer of WO_3_ nanopores. It was found that increasing the anodization time over 1 h does not lead to any significant change in the morphology of the prepared WO_3_ nanopores array.

The size of W nanopores could be easily tuned by changing the conditions of the anodic oxidation. The samples used for the SERS activity test were formed in 1 M H_2_SO_4_ solution 0.1 M NaF and with a potential equal to 40 V. The average porous size was estimated at 71.9 ± 14.6 nm. It was found that the porous size increases when increasing the voltage of the anodization process. When this process was carried out for 1 h with 30 V, the average porous size was 58.3 ± 11.2 nm, while in the case of 50 V for the same amount of time (1 h), the size was 86.2 nm ± 16.2. Figure 3 shows the morphology of the WO_3_ nanopores array formed by anodization of W foil at 1 M H_2_SO_4_ solution 0.1 M NaF for 1 h at various applied potentials. The presented data showed that in low potential, the formation of WO_3_ nanopores array cannot be completed, the structures still have defects, and the pore distribution is not homogenous. Samples prepared at higher potentials (40 and 50 V) do not vary between them.

The composition and concentrations of the electrolyte solution play a crucial role in the formation of WO_3_ nanopores array by the anodization process. Conducted research showed that NaF ions are necessary for the proper formation of WO_3_ nanopores. The minimal amount of F^-^ ions was specified at 0.002 M NaF in 100 mL of electrolyte. The data analysis showed that concentrations in the range of 0.05 to 0.15 M NaF do not lead to any significant change in the size of WO_3_ pores (Figure 4). All the tested anodization process conditions are shown in Table 1.

### 3.2. The XPS (X-ray Photoelectron Spectroscopy) Surface Characterization

The most intense peaks for W 4f region were assigned to the presence of W6+ (W4f_7/2_ = 35.3 ± 0.1 eV, Δ = 2.1 eV) [27,28] with a characteristic oxide loss structure set 5.5 eV above the W 4f_7/2_ peak (Figure 5). The peak set at lower binding energy (W4f_7/2_ = 33.3 ± 0.2 eV, Δ = 2.1 eV) according to literature might be related to W^4+^ [28,29,30,31], since W^5+^ is usually found around 34.3 ± 0.3 eV [28,31,32,33]. Based on the shape of the spectrum as well as on the fitting function, there is no evidence suggesting the existence of an additional tungsten oxidation state. The signal at O1s spectrum, located at 530.8 eV (O A), is typically assigned to lattice oxygen in WO_3_, whereas there is no consensus in the literature as to the assignment of the remaining, high-binding energy signals, especially since there is no information on C1s spectrum for some of them. Song Ling Wang et al. [28] ascribe peaks detected with a binding energy at 531.6 eV and 532.4 eV to the presence of adsorbed –OH originated from water and reduced tungsten oxidation states, respectively. L. Wang et al. [31] suggest that in the case of pristine WO_3_, the peak at 531.6 eV is presumably related to the surface contamination, whereas the same peak can be assigned to O^2−^ in the vicinity of oxygen vacancies. Therefore, it is suggested that the signals O B (532.4 eV) and O C (533.6 eV) originate from impurities containing groups such as C-O (286.0 eV on C 1 s) and O-C=O (288.8 eV on C 1 s). The C A signal in this case is most probably related to C-C and C-H bonds. Of course, it should be kept in mind that signals in the O1s spectrum may also overlap to some extent, hence the peak marked O B may also be a component of signals derived from O chemisorbed and O C from OH-/H_2_O. The XPS spectra of the F1s core electrons show no peak originating from fluoride-contacting species, which are commonly found in this kind of samples. 

The analyses of the survey spectrum suggest the following about the percentage composition of the surface: O: 27.51%, W: 8.35%, and C: 64.48%.

The common problem during the formation of WO_3_ nanostructures by the anodization method is the contamination of the surface by F^-^ ions which are intercalated into the nanopores structures. This problem is common in the case of organic electrolytes. One of the solutions leading to the elimination of F^-^ ions is the annealing process. In such cases, the anodized samples were heated in an oven at an elevated temperature in the range of 200–550 °C. However, it was found that temperatures higher than 500 °C led to structural damage to WO_3_ nanopores, what could be seen in Figure 6. In many cases, the lower temperature is not sufficient to completely remove F^-^ ions. However, the XPS analysis of formed samples does not reveal any presence of F^-^ ions. This effect is probably associated with the aqueous type of the applied electrolyte. The diffusion factor is higher because of the higher viscosity. In consequence, the F^-^ ions could be diffused from the surface and not intercalated into WO_3_ nanostructures.

### 3.3. Au Trisoctahedrons Characterization

The trisoctahedral gold nanoparticles were synthesised by a seed-mediated growth. The Au seeds were stabilized by CTAB and, in consequence, single-crystalline Au seeds were formed, in contrast to citrate-stabilized Au seeds, which are pentatwined. The single-crystalline Au seeds result in more favourable growth into anisotropic shapes. During the growth process, the CTAB was replaced by CTAC. With CTAB, the main product was identified as Au nanocubes. However, the reaction yield and stability are not as high as for Au trisoctahedrons. The Au NPs of bigger sizes were synthesised from the trisoctahedrons of smaller size. The considered shape could be understood as an octahedron with triangular pyramids on each side (Figure 7).

It should be noted that consistent growth of Au NPs does not change the shape and efficiency of the synthesis. Analysis of TEM (transmission electron microscopy) micrographs (Figure 8) showed that the average diameter (measured from edge to edge) is 70 ± 8 nm (62–95 nm) for the smallest NPs, then 93 ± 8 (74–120 nm) for the intermediates, and reach 114 ± 11 (84–135) for the biggest ones. The histograms were based on counting at least 100 nanoparticles for each sample. The small loss in homogeneity of size and shape distribution observed for the biggest Au NPs is caused by the few-steps growth process. Each growth step leads to some spread of size, therefore NPs formed in the third cycle exhibit slightly higher size distribution than others. The optical properties were investigated by UV-Vis spectrophotometry (see Figure 9). The red-shift of SPR (Surface Plasmon Resonance) could be observed associated with the increasing diameter of Au NPs. The 70 nm Au NPs exhibit SPR at 547 nm, the 94 nm Au NPs at 581 nm, while the biggest Au NPs at 610 nm. Moreover, also the half width increase with increasing size of Au NPs. This fact can indicate that the substantial growth of Au NPs leads to a decrease in size homogeneity in the sample. 

Figure 10 shows substrates with trisoctahedral gold nanoparticles. On the flat surface of WO_3_ (Figure 10a), the number of nanoparticles is small. On flat substrates, nanoparticles tend to accumulate at the edge of the droplet, which reduces the possible reinforcements obtained in the centre of the substrate. In the case of nanoparticles on porous substrates, we can observe that particles of sizes 70 and 94 nm can be locked on the pores.

### 3.4. SERS Measurements

R6G was used in the Raman studies. It is a dye often used as a probe molecule in SERS spectroscopy. Spectra of various concentrations of R6G in water were recorded for gold nanoparticles of sizes 71 nm, 94 nm, and 115 nm deposited on a nanostructured WO_3_ substrate (Figure 9). Characteristic bands were observed: 613 cm^−1^ corresponding to the C − C − C ring in-plane vibration mode, 772 cm^−1^ − C − H to the out-of-plane bend mode, 1126 cm^−1^ and 1185 cm^−1^ can be attributed to the N − H in-plane bend modes, while 1362 and 1510 cm^−1^ are derived from the N − H in-plane bend modes [34]. The highest intensity of spectra was obtained for nanoparticles with a size of 94 nm. It should be noted that in the case of nanoparticles of a size of 115 nm, there was a change in the band intensity ratio, especially visible for 1362 cm^−1^ and 1510 cm^−1^. This may be due to the different orientation of the molecule at 115 nm nanoparticles.

As can be seen in Figure 11, the lowest detection limit is reached for the Au NPs of diameter equal to 94 nm. This could result from the overlapping of the SPR band of Au NPs with the applied excitation wavelength. In this way, the generated local electromagnetic field is the highest. The smaller and bigger Au NPs exhibit lower detection limit towards R6G molecules. The SPR peak of 70 nm Au NPs is located at 547 nm. In the SERS experiments, the 632.8 nm laser line was used and hence those values do not overlap with the SPR of those Au NPs. In the case of the biggest Au NPs of 115 nm diameter, the SPR peak is widened, which can indicate a bigger size and shape dispersion, and hence lead to lower SERS enhancement. The reference SERS spectrum was collected on flat WO_3_ surfaces covered by 94 nm Au NPs.

Also, the recovery test was conducted for the proposed WO_3_-Au SERS platform. It was found that due to the porous structures of the substrate, simple water rinsing is not sufficient for analyte molecules washout, as the SERS signal from the analyte could be still observed. Also, the sonication procedure in water was applied. However, it was found that not just the analyte molecules were removed, but also some amount of Au NPs are detached, and in consequence, the SERS EF is significantly lower.

## 4. Conclusions

In this work, we successfully showed an efficient and repetitive method for WO_3_ nanopores array formation. Many factors affecting final morphology were carefully examined by SEM microscopy. Trisoctahedron Au NPs were successfully synthesised. It was proved that using smaller trisoctahedron Au NPs as seeds for growth of bigger nanoparticles permits easy tuning of the size of particles while maintaining the well-defined trisoctahedron shape. The UV-Vis spectroscopy showed shifting of SPR into longer wavelengths with successive growth of nanoparticles. The WO_3_—Au trisoctahedron modified nanoarray was successfully used as a SERS platform. The highest enhancement was observed for the Au NPs with 94 nm diameter.

## Figures and Tables

**Figure 1 materials-15-08706-f001:**
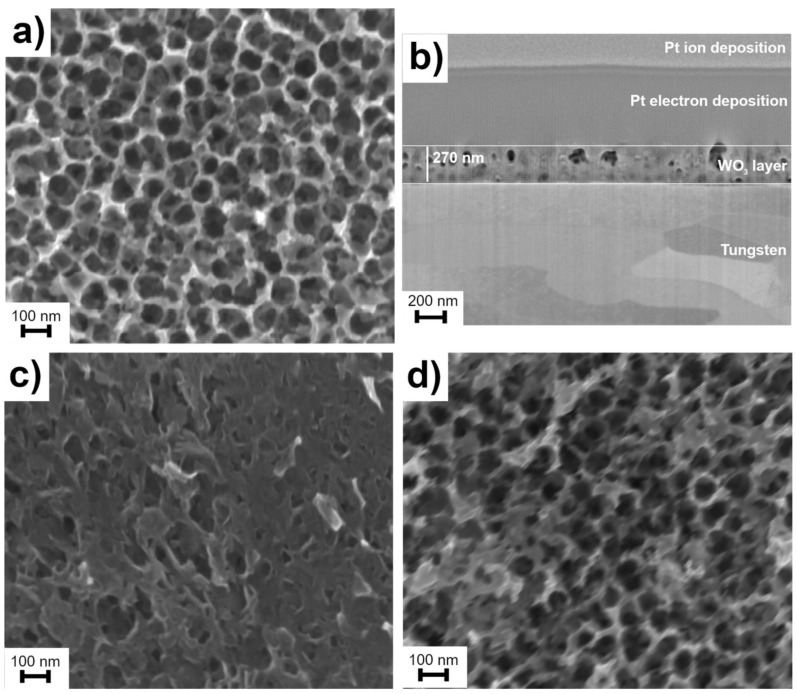
SEM micrographs of WO_3_ surfaces obtained using various electrolyte composition: (**a**) 0.1 M NaF in 1 M H_2_SO_4_, without protective layer; (**b**) FIB cross-section of (**a**) after deposition of Pt protective layer; (**c**) 0.1 M (NH4)_2_SO_4_ in water; and (**d**) 0.1 M NH_4_F in 1 M H_2_SO_4_.

**Figure 2 materials-15-08706-f002:**
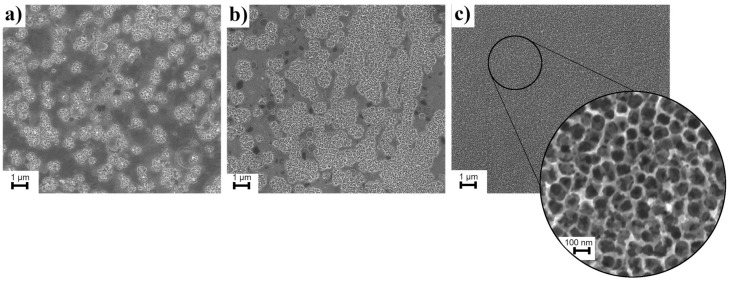
SEM micrographs showing the process of the WO_3_ nanopores formation into W foil during anodization in 1 M H_2_SO_4_ solution of 0.1 M NaF at various times at a constant potential equal to 40 V. (**a**) 15 min; (**b**) 30 min; (**c**) 60 min.

**Figure 3 materials-15-08706-f003:**
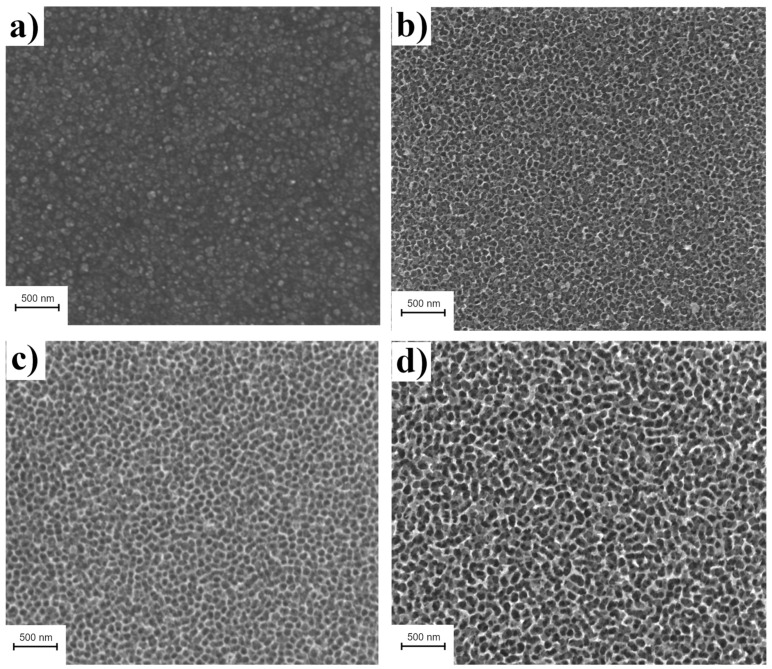
SEM micrographs of WO_3_ nanopores array formed by anodization of W foil in 1 M H_2_SO_4_ solution of 0.1 M NaF for 1 h at various potentials: (**a**) 20 V; (**b**) 30 V; (**c**) 40 V; and (**d**) 50 V.

**Figure 4 materials-15-08706-f004:**
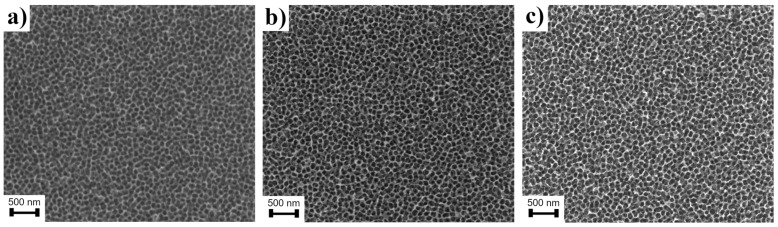
SEM micrographs showing the process of the formation of WO_3_ nanopores into W foil during anodization in 1 M H_2_SO_4_ solution containing various concentrations of NaF: (**a**) 0.05 M; (**b**) 0.1 M; and (**c**) 0.15 M.

**Figure 5 materials-15-08706-f005:**
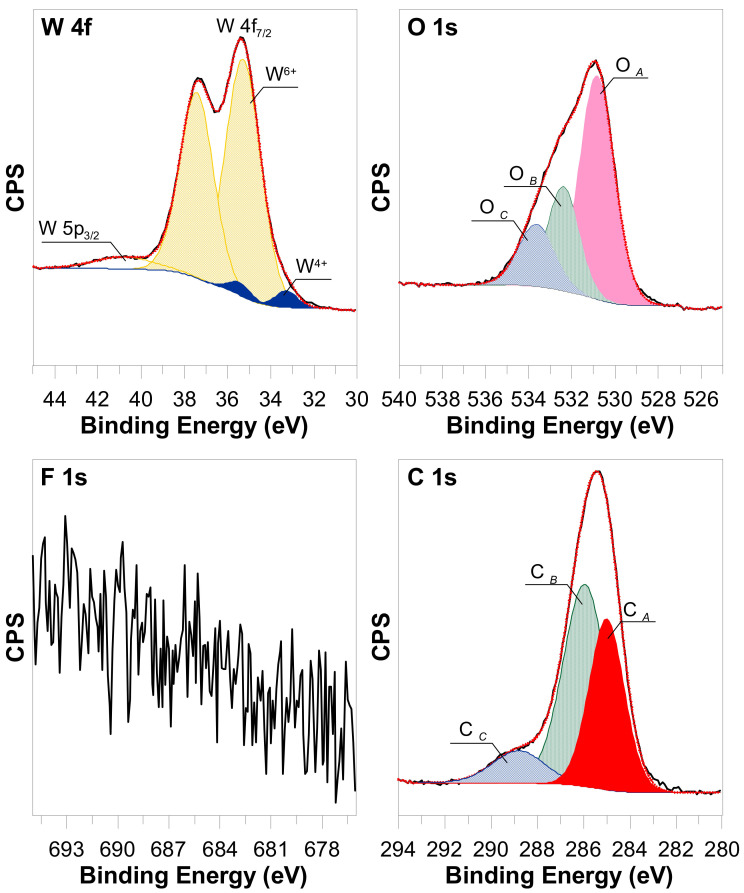
High resolution XPS spectra of WO_3_ nanopores array formed by an optimized anodization process.

**Figure 6 materials-15-08706-f006:**
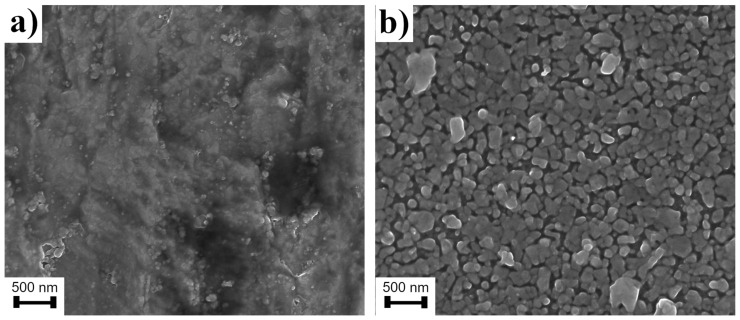
Surface of WO_3_ nanopores after annealing at: (**a**) 500 °C; (**b**) 550 °C.

**Figure 7 materials-15-08706-f007:**
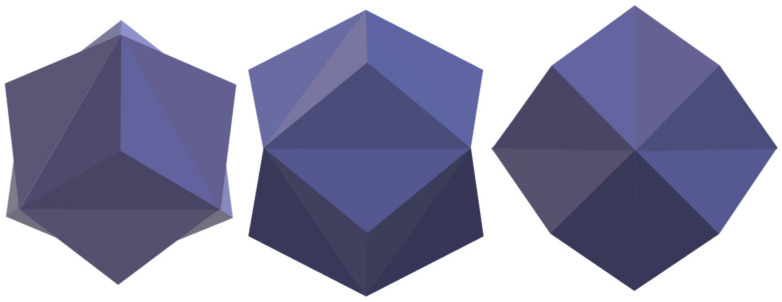
Model of trisoctahedron gold nanoparticles formed by the seed-mediated growth method.

**Figure 8 materials-15-08706-f008:**
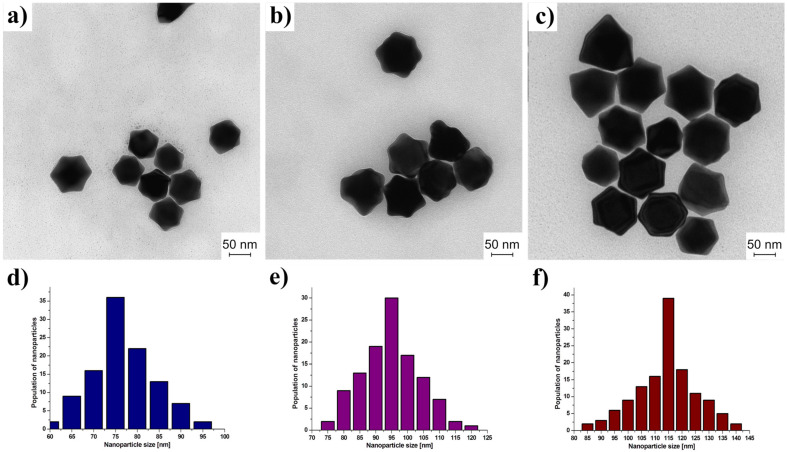
TEM micrographs of trisoctahedral gold nanoparticles of various diameters: (**a**) 70 nm; (**b**) 94 nm; and (**c**) 115 nm. Size distribution of nanoparticles: (**d**) 70 nm; (**e**) 94 nm; and (**f**) 115 nm.

**Figure 9 materials-15-08706-f009:**
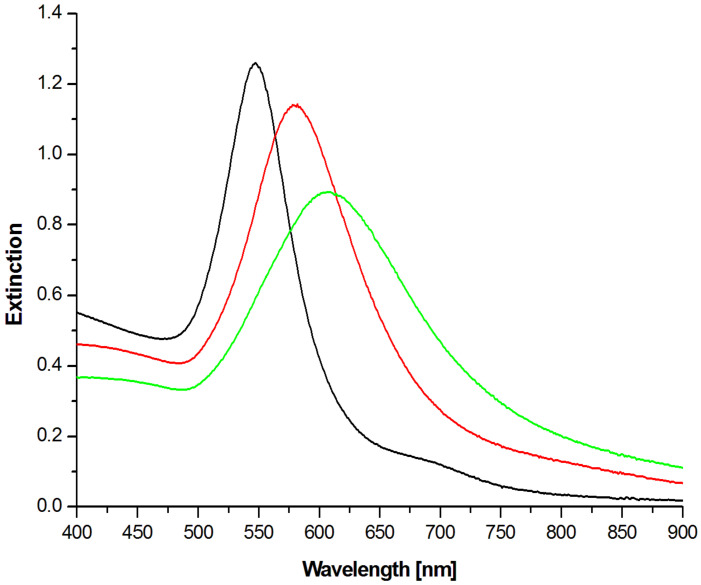
UV-Vis excitation spectra of trisoctahedral gold nanoparticles of various sizes: black—70 nm, red—94 nm, and green—115 nm.

**Figure 10 materials-15-08706-f010:**
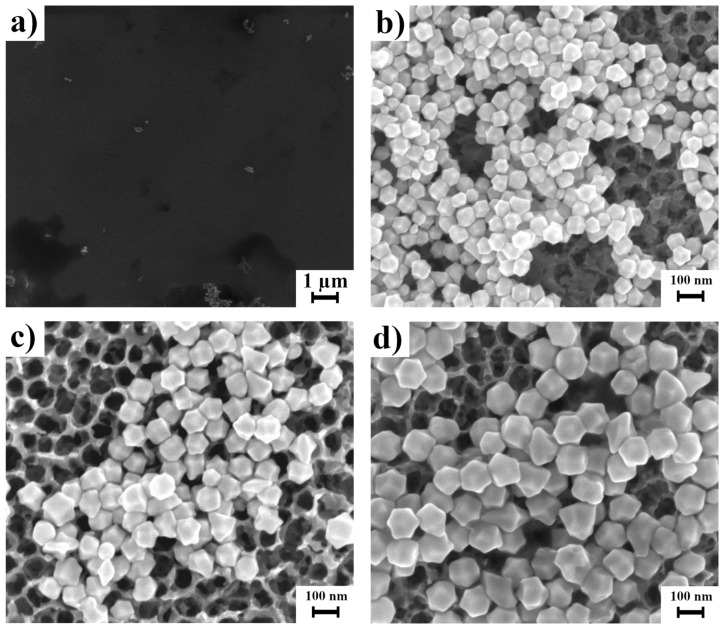
SEM micrographs showing the surface of flat WO_3_ covered by 94 nm trisoctahedral gold nanoparticles (**a**) and nanopores WO_3_ covered by trisoctahedral gold nanoparticles of 70 nm (**b**), 94 nm (**c**), and 115 nm (**d**).

**Figure 11 materials-15-08706-f011:**
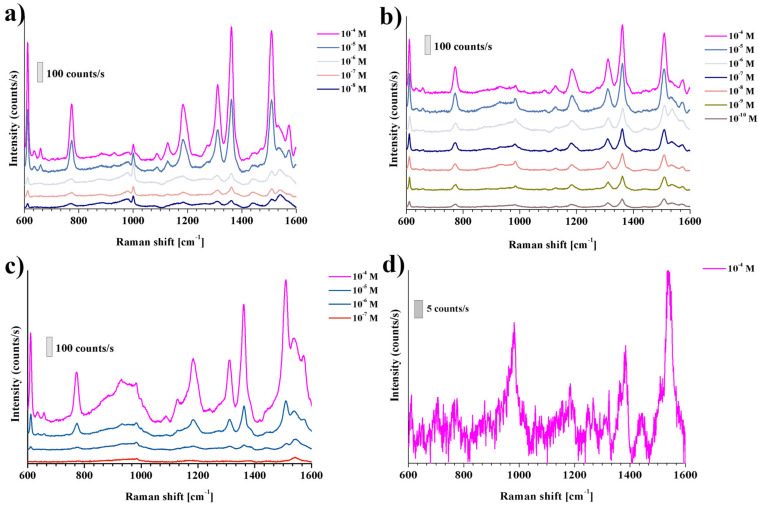
SERS spectra of R6G deposited on WO_3_ nanopores surfaces coated with Au trisoctahedral nanoparticles: (**a**) 71 nm; (**b**) 94 nm; and (**c**) 115 nm; (**d**) SERS spectrum of R6G deposited on flat WO_3_ coated with Au trisoctahedral nanoparticles (94 nm).

**Table 1 materials-15-08706-t001:** Detailed list of tested anodization conditions.

H_2_SO_4_	NaF	NH_4_F	(NH_4_)2SO_4_	Geometry of WO_3_
1 M	x	x	x	Flat
1 M	0.05 M	x	x	Nanopores array
1 M	0.1 M	x	x	Nanopores array
1 M	0.15 M	x	x	Nanopores array
1 M	x	0.1 M		Nanopores array (some deformations)
1 M	x	x	0.1 M	Rough surface
Results in optimized electrolyte composition: 1 M H_2_SO_4_, 0.1M NaF				
Time				
15 min	30 min	60 min		
Flat with holes	Flat with more holes	Nanopores		
Potential (average porous size)				
20 V	30 V	40 V	50 V	
No pores	Nanopores (58.3 nm)	Nanopores (71.9 nm)	Nanopores (86.2 nm)	

x—absence of reagent.

## Data Availability

Not applicable.

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
