# Peer review of "WO3 Nanopores Array Modified by Au Trisoctahedral NPs: Formation, Characterization and SERS Application"

_materials, 2022, doi:10.3390/ma15238706_

Round 1
Reviewer 1 Report
The authors did a good job on this manuscript. They investigated the WO3 nanopores array formation with various condition. They found efficient and repetitive method for synthesis, and did well comprehensive characterization. The techniques for characterization are well chosen, and well described.
The English is good, and I recommend publishing this paper.
Author Response
Thank you very much for your review.
Reviewer 2 Report
In the present manuscript, Jan Krajczewski et al. used the anodization method to form a nanoporous array of WO3. They characterized the nanoporous array by scanning electron microscopy (SEM) and X-ray photoelectron spectroscopy (XPS). The authors correlated the anodization parameters with the characterization results to identify the optimal parameters for obtaining an arrangement with homogeneous pore size and reproducibility. Subsequently, the authors synthesized trisoctahedral gold nanoparticles, which were deposited on their WO3 nanoporous arrays to explore the capabilities of this system as a SERS platform.
Certainly, the work of Jan Krajczewski and collaborators has a suitable experimental methodology and is engaging. In this type of experimental work, it is essential to describe the experimental procedure and its results accurately, something that the authors DID do and that significantly facilitates the reading of the complete work.
I believe the work can be published in the MDPI Materials Journal if the following adjustments are made (below, I list the requested corrections in the order in which they appeared as I read the manuscript. Obviously, some are more important than others. I ask authors to be much more thorough in those that they identify are of greater relevance to improve their work).
1. Figure 1.b shows us a cross-section of the nanoporous layer of WO3. However, this layer seems to be between two other layers, the Pt electron deposition, and the tungsten. If the above is correct, how can the surface image of figure 1.a be observed? i.e., was there any special preparation of the sample? If this is not the case, could you explain Figures 1.a and 1.b in greater depth and clarity?
2. The authors mention that by varying the amount of F- ions, through varying NaF from 0.25g to 0.75g, no significant change in the size of the WO3 nanopores is generated. However, for 0.5g of NaF, what appear to be "voids" or some structure are observed in the micrograph. The micrograph is NOT filled with nanopores, as it IS with 0.25 and 0.75 g of NaF. Could the authors give an explanation as to why this happens?
3. In line 269, it is discussed that exposing nanoporous arrays to annealing temperatures higher than 500°C generates structural damage in the nanopores of WO3. Could the authors elaborate on what they think is the mechanism by which this temperature causes structural damage, and not 300°C, to mention just one example?
4. Regarding the previous question. It would be convenient if the authors reported some micrographs of the nanoporous arrangements of WO3 with annealing above 500°C so that the reader can identify the mentioned structural damages.
5. In line 291, it is mentioned that the increase in AuNPs leads to a decrease in the size homogeneity of the synthesized AuNPs. It would be convenient if the authors offered a more detailed explanation of why (that is, why there is a trade-off between increasing the size of the AuNPs and decreasing size homogeneity).
6. The authors mention that the SERS platform they propose is composed of their porous arrangement of WO3 "decorated" with the AuNPs. They must present an SEM micrograph of said arrangement to identify its nanostructural characteristics.
7. The authors test R6G on their WO3 nanoporous platforms "decorated" with AuNPs. However, they need a control sample. That is, they need to test R6G on substrates (glass, to give an example) that are "added or decorated" with AuNPs; or on some nanoporous array of WO3 WITHOUT the "ideal" porosity shown in fig 1.a, 2.c or 3.c with the aim that the reader clearly identifies the influence of nanoporous WO3 in combination with the AuNPs proposed in this article.
8. Finally, the following minor errors must be taken care of:
a. Enhancement Factor (EF) of line 29 is not previously defined.
b. The same goes for the Limit of Detection (LOD) on line 45.
c. The same thing happens on line 164 with EG and DEG.
d. In line 198, the phrase (Figure 2) should not be between two parentheses. Figure 2 could instead be the beginning of the sentence.
e. The same thing happens on line 215.
f. In the captions of Figure 6, it would be convenient for the authors to expand the description more. With the aim that readers do NOT have to refer to the main text to know what the figure means.
g. In line 286, the phrase "for the another" is not the most appropriate. Authors can use "for the intermediates," for example.
Author Response
In the present manuscript, Jan Krajczewski et al. used the anodization method to form a nanoporous array of WO3. They characterized the nanoporous array by scanning electron microscopy (SEM) and X-ray photoelectron spectroscopy (XPS). The authors correlated the anodization parameters with the characterization results to identify the optimal parameters for obtaining an arrangement with homogeneous pore size and reproducibility. Subsequently, the authors synthesized trisoctahedral gold nanoparticles, which were deposited on their WO3 nanoporous arrays to explore the capabilities of this system as a SERS platform. Certainly, the work of Jan Krajczewski and collaborators has a suitable experimental methodology and is engaging. In this type of experimental work, it is essential to describe the experimental procedure and its results accurately, something that the authors DID do and that significantly facilitates the reading of the complete work.
I believe the work can be published in the MDPI Materials Journal if the following adjustments are made (below, I list the requested corrections in the order in which they appeared as I read the manuscript. Obviously, some are more important than others. I ask authors to be much more thorough in those that they identify are of greater relevance to improve their work).
- Figure 1.b shows us a cross-section of the nanoporous layer of WO3. However, this layer seems to be between two other layers, the Pt electron deposition, and the tungsten. If the above is correct, how can the surface image of figure 1.a be observed? i.e., was there any special preparation of the sample? If this is not the case, could you explain Figures 1.a and 1.b in greater depth and clarity?
The Pt deposition is only applied for the FIB characterized samples, in all other samples the WO3 surfaces were not covered by anything.
- The authors mention that by varying the amount of F- ions, through varying NaF from 0.25g to 0.75g, no significant change in the size of the WO3 nanopores is generated. However, for 0.5g of NaF, what appear to be "voids" or some structure are observed in the micrograph. The micrograph is NOT filled with nanopores, as it IS with 0.25 and 0.75 g of NaF. Could the authors give an explanation as to why this happens?
Image shown in the micrograph was inappropriate. The correct micrograph is shown in the revised version
- In line 269, it is discussed that exposing nanoporous arrays to annealing temperatures higher than 500°C generates structural damage in the nanopores of WO3. Could the authors elaborate on what they think is the mechanism by which this temperature causes structural damage, and not 300°C, to mention just one example?
Thank you for this point, in a revised version of the manuscript we added the SEM images of annealed samples (fig. 6), as well as commented on this effect in the main text.
- Regarding the previous question. It would be convenient if the authors reported some micrographs of the nanoporous arrangements of WO3 with annealing above 500°C so that the reader can identify the mentioned structural damages.
Thank you for this point, in a revised version of the manuscript we added the SEM images of annealed samples at 500oC and 550oC(fig. 6).
- In line 291, it is mentioned that the increase in AuNPs leads to a decrease in the size homogeneity of the synthesized AuNPs. It would be convenient if the authors offered a more detailed explanation of why (that is, why there is a trade-off between increasing the size of the AuNPs and decreasing size homogeneity).
In the corrected version of the manuscript, we described the seed-mediated growth process with more details and a short discussion about the reason for losing some homogeneity in size and shape.
- The authors mention that the SERS platform they propose is composed of their porous arrangement of WO3 "decorated" with the AuNPs. They must present an SEM micrograph of said arrangement to identify its nanostructural characteristics.
Thank you very much, indeed the SEM micrographs of the WO3 nanopores array decorated by Au trisoctahedral NPs are crucial for this paper. In the revised version of the manuscript, we added four SEM micrographs showing Au NPs distribution on flat and nanopore WO3 surfaces covered by Au NPs with various sizes.
- The authors test R6G on their WO3 nanoporous platforms "decorated" with AuNPs. However, they need a control sample. That is, they need to test R6G on substrates (glass, to give an example) that are "added or decorated" with AuNPs; or on some nanoporous array of WO3 WITHOUT the "ideal" porosity shown in fig 1.a, 2.c or 3.c with the aim that the reader clearly identifies the influence of nanoporous WO3 in combination with the AuNPs proposed in this article.
Thank you for this comment, we introduced a R6G SERS spectra recorded on a flat WO3 surface covered by Au NPs as reference. As could be expected, the recorded SERS signal is significantly lower and very randomly distributed around the sample surface in comparison with the porous WO3 surface.
- Finally, the following minor errors must be taken care of:
- Enhancement Factor (EF) of line 29 is not previously defined.
- The same goes for the Limit of Detection (LOD) on line 45.
- The same thing happens on line 164 with EG and DEG.
- In line 198, the phrase (Figure 2) should not be between two parentheses. Figure 2 could instead be the beginning of the sentence.
- The same thing happens on line 215.
- In the captions of Figure 6, it would be convenient for the authors to expand the description more. With the aim that readers do NOT have to refer to the main text to know what the figure means.
Thank you for this comment, all abbreviations were defined while used for the first time.
- In line 286, the phrase "for the another" is not the most appropriate. Authors can use "for the intermediates," for example.
Thank you, we changed this phase as you recommended.
Reviewer 3 Report
In this paper, Authors propose WO3 nanopores array modified by AuNPs. The topic is of significant technological impact. However, some improvements are necessary before publication. An itemized list of comments is reported below:
- The Introduction section needs to be more focused on the advancement of this technique in respect to the state of the art.
- Section 2.1: details about a lot of chemicals are missing: HAuCl4, CTAB, CTAC, NaBH4, NaOH, ascorbic acid, etc.. Please add.
- All the reported experimental protocols are expressed as weights of reactants in various volumes of solutions. I warmly suggest to use concentrations for an univocal identification of reaction conditions.
- It is not clear why, during the synthesis of AuNPs the counterion of cetyltrimethylammonium is changed (bromide of CTAB/chloride of CTAC). Please explain.
- Lines 102-103: “For SERS measurements […] more centrifugation cycle”. Do you mean that you added a lower solution volume after centrifugation? So, which is the final concentration used?
- It is not correct to use a “smart function” for elemental quantification and a “Shirley function” for curve fitting, during background subtraction. It would be better to use the same background subtraction function.
- Please add a table summarizing all the combination of experimental conditions used for WO3 and AuNPs.
- Line 145: reaction is not balanced in charges, please revise.
- Lines 162-168: please add references.
- Line 177: “in the presence of NaF and NH4F in a solution of…”. Are you referring to 0.5 g of these reactants?
- Please explain the difference between the “Pt ion deposition” and the “Pt electron deposition” reported in Fig. 1b.
- Lines 209-220: please add errors to the average pores sizes.
- Authors use “WOx” and “WO3” without distinction. Indeed, they are not the same compound, and the correct name should be chosen and used.
- Errors on XPS peak positions should be added.
- A C% of 70% is too high to be “originated from impurities”. In case of adventitious contamination, it should not be more than 20-25%. Hence, Authors should argue on the origin of this organic layer covering the WO3 surface.
- Where is Pt, clearly visible in Fig. 1b?
- Please add a table with complete surface atomic composition.
- A peak seems to be missing in Fig. 5, for the O1s spectrum, at about 528-529 eV. Maybe background subtraction needs to be revised.
- Section 3.3: average sizes for AuNPs must be reported with the correct number of significant digits, i.e., 71 ± 8 nm, 94 ± 8 nm, etc..
- Fig. 7d is poorly readable and should be separated in 3 histograms, with corresponding fitting.
- Please add SEM characterization of Au-modified WO3 substrate for SERS measurements.
- Which is the durability of the proposed substrate? Can it be washed/reused?
Minor remarks:
- Please check for subscripts and superscripts throughout the paper: most of them are missing, starting from the title.
- ALL acronyms (SEM, XPS, TEM, SERS, EF, EG, DEG, NTs, etc.) should be properly explicated in the manuscript: although common to a materials science audience, the paper should be easily readable by everyone.
- Line 2: change “modifed” with “modified”.
- Line 65: delete double dot.
- Line 89: change “was” with “were”.
- Line 196: change “NH4SO4” with “(NH4)2SO4”.
- Line 198: remove unnecessary dot.
Author Response
In this paper, Authors propose WO3 nanopores array modified by AuNPs. The topic is of significant technological impact. However, some improvements are necessary before publication. An itemized list of comments is reported below:
- The Introduction section needs to be more focused on the advancement of this technique in respect to the state of the art.
The proper sentences were added to the introduction section
- Section 2.1: details about a lot of chemicals are missing: HAuCl4, CTAB, CTAC, NaBH4, NaOH, ascorbic acid, etc.. Please add.
Thank you for your comment, in the revised version we introduce all missing chemical compounds and delivery sources.
- All the reported experimental protocols are expressed as weights of reactants in various volumes of solutions. I warmly suggest to use concentrations for an univocal identification of reaction conditions.
Good point, we recalculated all weights concentrations into molar concentrations for more clarity.
- It is not clear why, during the synthesis of AuNPs the counterion of cetyltrimethylammonium is changed (bromide of CTAB/chloride of CTAC). Please explain.
Thank you for this comment, in the revised version we explain the reason for changing the stabilizing agent during the Au NPs synthesis.
- Lines 102-103: “For SERS measurements […] more centrifugation cycle”. Do you mean that you added a lower solution volume after centrifugation? So, which is the final concentration used?
Thank you for this comment, we changed this part of the text in the more clear way.
- It is not correct to use a “smart function” for elemental quantification and a “Shirley function” for curve fitting, during background subtraction. It would be better to use the same background subtraction function.
The authors agree with this suggestion and would like to assure that the spectra were reprocessed according to the standard method. The information about the usage of a “smart function” should not appear in the methodology description. Authors changed the text:
The chemical states of individual elements were verified by X-ray photoelectron spectroscopy (XPS) using a Microlab 350 (Thermo Electron, East Grinstead, UK) spectrometer. For this purpose, the X ray excitation source (AlKα anode: power 300 W, voltage 15 kV, beam current 20 mA) was used. The lateral resolution of XPS analysis was 130 about 0.2 cm2. The high-resolution XPS spectra were recorded using the following parameters: pass energy 40 eV, energy step size 0.1 eV. A smart function of background subtraction was used to obtain the XPS signal intensity. XPS spectra were reprocessed using CasaXPS (2.3.18PR1.0) software. Spectra were
fitted with GL(30) line shape after Shirley background subtraction, and subsequently charge corrected to give a C 1s at 285 eV.
- Please add a table summarizing all the combination of experimental conditions used for WO3 and AuNPs.
In the revised version of the manuscript, we added the table summarizing all the combinations of WO3 synthesis.
- Line 145: reaction is not balanced in charges, please revise.
Thank you for this comment, we corrected the charges in the reaction.
- Lines 162-168: please add references.
The proper references were added.
- Line 177: “in the presence of NaF and NH4F in a solution of…”. Are you referring to 0.5 g of these reactants?
Thank you for this point, in a revised version of the manuscript we changed the Figure description for more clarity.
- Please explain the difference between the “Pt ion deposition” and the “Pt electron deposition” reported in Fig. 1b.
When depositing Pt on a sensitive surface it is very important to overcome effects that may destroy the structure of the surface. It is highly recommended to build up a layer of at least 100 nm and then continue with an ion beam that is faster but needs higher currents. Using electron beam-induced deposition fewer Ga+ is introduced to the material, which is seen as a difference in contrast on SEM images. To the methodology part the FIB studies description was added:
Material cross-section was performed using electron-ion (Ga+) scanning Crossbeam 540X microscope. Prior the cut, the sensitive to the ion beam surface was protected with electron-beam induced Pt deposition (2 kv, 4 nA) and after that with ion- beam induced Pt deposition (30 kV, 300 pA). The cross-section was performed with 30 kV and 3 nA, whereas the final polishing was stopped at 30 kV and 700 pA.
- Lines 209-220: please add errors to the average pores sizes.
Thank you, we added the errors for pores size diameter.
- Authors use “WOx” and “WO3” without distinction. Indeed, they are not the same compound, and the correct name should be chosen and used.
Thank you for this comment, we changed all WOx on WO3.
- Errors on XPS peak positions should be added.
This is a very good point. Usually, authors add this information, however, in this situation, the presented XPS studies are just a part of the whole experiment. Therefore in order to compare all obtained spectra authors performed deconvolution using constrained peaks position for all regions. In this situation, the shifts on BE scale are 0 eV.
- A C% of 70% is too high to be “originated from impurities”. In case of adventitious contamination, it should not be more than 20-25%. Hence, Authors should argue on the origin of this organic layer covering the WO3 surface.
The authors agree with this comment and were aware of this problem from the beginning. Analyzing the process of WO3 formation there are no C-containing reagents. The surface is highly porous therefore carbon species can originate from the plastic box in which the sample was stored, from the atmosphere (CHX, C-O) or even they could be adsorbed from the load lock chamber of the XPS apparatus. The only way to understand the presence is the preparation of a new sample and measurement of the fresh surface without its long exposition to the air. If C- containing species would be still present the sample could be exposed to O2 at 250C in the flow reactor chamber in order to clean the surface. To do so authors need more time for this kind of experiment, whereas in the authors' opinion the origination of carbon species is not the most important part of this work. However, authors reprocessed background subtraction of the survey spectrum obtaining the following atomic concentrations:
O 1s 27.51
W 4f 8.35
C 1s A 26.64
C 1s B 10.31
C 1s C 27.19
The whole C 1s at concentration is equal around 64.5%.
- Where is Pt, clearly visible in Fig. 1b?
We added a proper captions on the Fig. 1b for more clarity.
- Please add a table with complete surface atomic composition.
Instead of a table Authors added the text:
The analyse of the survey spectrum suggest the following at percentage composition of the surface: O: 27,51, W: 8.35 and C: 64,48%.
- A peak seems to be missing in Fig. 5, for the O1s spectrum, at about 528-529 eV. Maybe background subtraction needs to be revised.
According to the Reviewer’s suggestion, the background subtraction has been revised since there are no indications suggesting the existence of another lattice oxygen component.
- Section 3.3: average sizes for AuNPs must be reported with the correct number of significant digits, i.e., 71 ± 8 nm, 94 ± 8 nm, etc..
Thank you for your comment, we corrected the average size of Au NPs.
- Fig. 7d is poorly readable and should be separated in 3 histograms, with corresponding fitting.
Thank you for this point, we changed Figure 7d.
- Please add SEM characterization of Au-modified WO3 substrate for SERS measurements.
Thank you very much, indeed the SEM micrographs of the WO3 nanopores array decorated by Au trisoctahedral NPs are crucial for this paper. In the revised version of the manuscript, we added four SEM micrographs showing Au NPs distribution on flat and nanopore WO3 surfaces covered by Au NPs with various sizes.
- Which is the durability of the proposed substrate? Can it be washed/reused?
Thank you for this comment, in the revised manuscript version we shortly discussed the reusability of the proposed SERS platform.
Minor remarks:
- Please check for subscripts and superscripts throughout the paper: most of them are missing, starting from the title.
We corrected all subscripts and superscripts in the manuscript.
- ALL acronyms (SEM, XPS, TEM, SERS, EF, EG, DEG, NTs, etc.) should be properly explicated in the manuscript: although common to a materials science audience, the paper should be easily readable by everyone.
Thank you for these comments, all abbreviations were defined while used for the first time.
- Line 2: change “modifed” with “modified”.
- Line 65: delete double dot.
- Line 89: change “was” with “were”.
- Line 196: change “NH4SO4” with “(NH4)2SO4”.
- Line 198: remove unnecessary dot.
Thank you for your comments, we corrected all typos.
Reviewer 4 Report
[1] spelling check should be done for words like “anodization” in line 9 and many others in the manuscript.
[2] What is EF? This abbreviation is used continuously without giving the full name before abbreviating. The authors should give a full name before abbreviating.
[3] The authors should add more keywords, at least 5 should be given as per the general journal requirements.
[4] Line 284-286 “Analysis of TEM micrographs (Figure 7) showed that average diameter (measured from edge to edge) is 70.6±8.2nm for the smallest, then 93.9±8.0 for the another, and reach 114.7±11.1 for the biggest ones. The histograms was based on counting at least 100 nanoparticles for each sample”. The average diameter of the smallest Au nanoparticles is approximately 63 nm and the largest is 135 nm as per the histogram displayed. Thus, 70.6 and 114 nm cannot be the smallest and largest sizes, respectively. The authors need to address this.
[5] What happens to the nanostructured WO3 surface after Au was deposited on the surface? The SEM images of the WO3 - Au trisoctahedron modified nanoarray are needed to discuss what happens to the pore structures after the deposition of Au and what impact could that have on the SERS application.
[6] Many grammatical flaws in the manuscript makes it hard for one to read the manuscript, thus I suggest that the authors send their manuscript for English editing to ensure that their wonderful work is clearly presented.
[7] The labelling on the y-axis of figure 8 should be excitation, not extinction. This clearly shows that English editing might help the authors.
[8] The y-axis of figure 9 is not labelled.
Author Response
[1] spelling check should be done for words like “anodization” in line 9 and many others in the manuscript.
Thank you for this comment, we corrected all typos.
[2] What is EF? This abbreviation is used continuously without giving the full name before abbreviating. The authors should give a full name before abbreviating.
We deciphered the EF abbreviation when using it for the first time.
[3] The authors should add more keywords, at least 5 should be given as per the general journal requirements.
Thank you for this comment, we had added more keywords.
[4] Line 284-286 “Analysis of TEM micrographs (Figure 7) showed that average diameter (measured from edge to edge) is 70.6±8.2nm for the smallest, then 93.9±8.0 for the another, and reach 114.7±11.1 for the biggest ones. The histograms was based on counting at least 100 nanoparticles for each sample”. The average diameter of the smallest Au nanoparticles is approximately 63 nm and the largest is 135 nm as per the histogram displayed. Thus, 70.6 and 114 nm cannot be the smallest and largest sizes, respectively. The authors need to address this.
Thank you for this comment, we changed the text in this section for more clarity.
[5] What happens to the nanostructured WO3 surface after Au was deposited on the surface? The SEM images of the WO3 - Au trisoctahedron modified nanoarray are needed to discuss what happens to the pore structures after the deposition of Au and what impact could that have on the SERS application.
Thank you very much, indeed the SEM micrographs of the WO3 nanopores array decorated by Au trisoctahedral NPs are crucial for this paper. In the revised version of the manuscript, we added four SEM micrographs showing Au NPs distribution on flat and nanopore WO3 surfaces covered by Au NPs with various sizes.
[6] Many grammatical flaws in the manuscript makes it hard for one to read the manuscript, thus I suggest that the authors send their manuscript for English editing to ensure that their wonderful work is clearly presented.
In the revised version of the manuscript, we tried to correct all grammatical errors.
[7] The labelling on the y-axis of figure 8 should be excitation, not extinction. This clearly shows that English editing might help the authors.
We measured the Au NPs suspension by transmittance UV-VIS spectroscopy. As Extinction we understand the total light loss caused by both absorption and scattering, therefore in our opinion, the y-axis in figure 8 is correct.
[8] The y-axis of figure 9 is not labelled.
Thank you, in a revised version of the manuscript we had to add the y-axis description.
Reviewer 5 Report
A manuscript contain original research work, where the authors have presented the results of synthesis of WO3 nanoarrays by anodizing tungsten foil in aqueous medium containing fluoride ions, as well as using WO3-Au modified nanoarrays as an active SERS platform. The article is very intersting for researchers using the SERS method. I recommend accepting current manuscript, however there are some irregularities in this manuscript that should be corrected:
1. Line 23 - the SERS abbreviation should be deciphered
2. Line 109 - the R6G abbreviation should be deciphered and remove on line 304
3. Line 288 - the SPR abbreviation should be deciphered
4. Line 344 - at the end of phrase you have to add the word "sizes or diameters"
5. Line 366 - typo, juging by DOI, year 2022
Author Response
A manuscript contain original research work, where the authors have presented the results of synthesis of WO3 nanoarrays by anodizing tungsten foil in aqueous medium containing fluoride ions, as well as using WO3-Au modified nanoarrays as an active SERS platform. The article is very interesting for researchers using the SERS method. I recommend accepting current manuscript, however there are some irregularities in this manuscript that should be corrected:
- Line 23 - the SERS abbreviation should be deciphered
- Line 109 - the R6G abbreviation should be deciphered and remove on line 304
- Line 288 - the SPR abbreviation should be deciphered
Thank you for this comment, we deciphered all abbreviations when using them for the first time.
- Line 344 - at the end of phrase you have to add the word "sizes or diameters"
So sorry, we could not find the proper phrase in line 344 and in any other when the proposed text is missing.
- Line 366 - typo, juging by DOI, year 2022
The cited manuscript was published in 2022, but the Journals give them the next year's publish date.
Reviewer 6 Report
Jan et al. present an WO3 nanopores array modifed by Au trisoctahedral NPs: for mation, characterization and SERS application. The issue studied here is regarding to develop s synthesis parameterization of a well ordered WO3 nanopores array by anodization method in aqueous solution containing F- ions. The comparative novelty of this work is to get a better understanding of shape and size of Au NPs was analysed by TEM microscopy and WO3-Au platform showing a SERS activity. In my opinion, this work is of interest to researchers in the field of WO3 nanostructures in plasmonics
The authors should consider the following comments to improve their manuscript.
MA1. The Introduction must be improved by incorporating more recent references including methods for WO3 nanostructures and determination of the NPs properties.
MA2. The workflow of the present work in Figure should be addressed more concisely and in details.
MA3. Please merge and explain Figure 1 and Figure 2 in manuscript. And please address concise data and explain details in the revised manuscript.
MA4. Please merge and explain Figure 3 and Figure 4 in manuscript. And please address concise data and explain details in the revised manuscript.
MA5. In Figure 6, please address concise data of trisoctahedron model and explain the evidence or clue details in the revised manuscript.
MA6. In conclusion, please the contents detailed should be addressed including future scope and applications for a better understanding of efficient and repetitive method for WO3 nanopores array formation.
The subject may be interesting enough WO3 nanostructures but only after major, deep revision, if at all possible, to resolve the above.
Author Response
Jan et al. present an WO3 nanopores array modifed by Au trisoctahedral NPs: for mation, characterization and SERS application. The issue studied here is regarding to develop s synthesis parameterization of a well ordered WO3 nanopores array by anodization method in aqueous solution containing F- ions. The comparative novelty of this work is to get a better understanding of shape and size of Au NPs was analysed by TEM microscopy and WO3-Au platform showing a SERS activity. In my opinion, this work is of interest to researchers in the field of WO3 nanostructures in plasmonics
The authors should consider the following comments to improve their manuscript.
MA1. The Introduction must be improved by incorporating more recent references including methods for WO3 nanostructures and determination of the NPs properties.
Thank you for this comment, we added more references including methods for WO3 nanostructures synthesis by anodization method.
MA2. The workflow of the present work in Figure should be addressed more concisely and in details.
We changed Figures' captions where necessary.
MA3. Please merge and explain Figure 1 and Figure 2 in manuscript. And please address concise data and explain details in the revised manuscript.
MA4. Please merge and explain Figure 3 and Figure 4 in manuscript. And please address concise data and explain details in the revised manuscript.
Unfortunately, we are not able to introduce your comment in a revised version of the manuscript. The mentioned Figures showed SEM micrographs of samples formed in different conditions, therefore for more clarity, we put them separately. All figures are mentioned in the main text of the manuscript.
MA5. In Figure 6, please address concise data of trisoctahedron model and explain the evidence or clue details in the revised manuscript.
Thank you for this comment, the synthesis of Au trisoctahedron nanoparticles was based on cited manuscript. Both, TEM and SEM as well as cited manuscript analysis presented in our work confirmed the formed shape of nanoparticles.
MA6. In conclusion, please the contents detailed should be addressed including future scope and applications for a better understanding of efficient and repetitive method for WO3 nanopores array formation.
We mentioned in the introduction part the main application of WO3 nanostructures nowadays, and shortly describe the future scope and application of the material.
The subject may be interesting enough WO3 nanostructures but only after major, deep revision, if at all possible, to resolve the above.
Round 2
Reviewer 2 Report
The authors have made extensive corrections to their manuscript, and it has improved considerably. The manuscript can be published in its current version.
Reviewer 3 Report
Authors have properly addressed all Reviewer's comments, and the paper is now suitable for publication.
Reviewer 6 Report
The authors have improved the manuscript for publication.